

# Air Mass Origin Effects on Antarctic Snow Isotopic Composition: An Observation and Modelling Study

Agnese Petteni[1,2], Mathieu Casado[2], Christophe Leroy-Dos Santos[2], Amaelle Landais[2], Niels Dutrievoz[2], Cécile Agosta[2], Pete D. Akers[3,4], Joel Savarino[4], Andrea Spolaor[5], Massimo Frezzotti[6] and Barbara Stenni[1]

[1]Ca' Foscari of Venice, Department of Environmental Sciences, Informatics and Statistics, Mestre (Venice), Italy
[2]LSCE/IPSL, CEA-CNRS-UVSQ, Université Paris-Saclay, Gif-sur-Yvette, France
[3]Geography, School of Natural Sciences, Trinity College Dublin, Ireland
[4]Université Grenoble Alpes, CNRS, IRD, Grenoble INP, INRAE, IGE, F-38000 Grenoble, France
[5]Institute of Polar Sciences, National Research Council of Italy, Venice, Italy
[6]Roma Tre University, Department of Science, Rome, Italy

*Correspondence to*: Agnese Petteni (agnese.petteni@unive.it)

**Abstract.**

Water stable isotopes ($\delta^{18}$O and $\delta$D) from ice cores are commonly used to reconstruct past temperature variations because of their well-established relationship with local air temperature. However, depositional and post-depositional effects lead to large uncertainties to use this proxy in Antarctica. Depositional effects are largely influenced by the origin of precipitation moisture, which exhibits asymmetries shaped by the continent's geographical and topographical features. Additionally, precipitation intermittency - especially in low-accumulation areas – introduce aliasing in the recorded signal, significantly limiting the temperature signal that can be retrieved. Post-depositional processes, such as sublimation and firn-atmosphere exchange, can further alter the isotopic composition of snow before its transformation into ice, potentially modifying the correlation between $\delta^{18}$O and air temperature for snow samples. Here, we present new water isotope measurements from surface snow collected during the East Antarctic International Ice Sheet Traverse (EAIIST) across a remote region of the East Antarctic plateau. The traverse - crossing a transitional zone between predominately Indian and Pacific moisture sources - provides direct insights into the key role of air mass origin in shaping the $\delta^{18}$O-temperature relationship. A comparison between snow isotopic values and precipitation simulations from the atmospheric general circulation model LMDZ6iso shows that the model accurately captures the spatial variation of the $\delta^{18}$O–temperature relationship observed in snow. This result also supports the model's ability to predict the temporal slope required to calibrate isotopic ice core records for past temperature reconstructions, even in regions where precipitation events originate from different sources. Finally, the impact of sublimation on $\delta^{18}$O and *d-excess* (an effect that must be considered for accurate paleoclimatic reconstructions) is evidenced for the region covered by EAIIST.



## 1. Introduction

Ice cores are valuable archives of past climatic and environmental conditions through many environmental proxies both trapped in the ice, such as air bubbles and aerosols, as well as the ice itself, such as its water isotopic composition. In East Antarctica, water stable isotopes ($\delta^{18}$O and $\delta$D) from the deepest ice cores have been used to reconstruct past temperatures extending back
800,000 years (EPICA community members et al., 2004; Jouzel, 2007), and recent efforts aim to extend these reconstructions to 1.5 million years (Beyond EPICA Oldest Ice Core - Parrenin et al., 2017; Lilien et al., 2021). In this region, ice core temporal resolution is typically limited to multi-annual to multi-decadal scale (Ekaykin et al., 2002; Baroni et al., 2011; Münch et al., 2016). This limitation is due to low accumulation rates and intermittency of precipitation which cause aliasing of the climatic signal recorded by sporadic snowfall events (Ekaykin et al., 2016; Laepple et al., 2018; Casado et al., 2020; Münch et al.,
2021). However, ice cores in high-accumulation coastal regions of Antarctica cover shorter temporal ranges but provide finer temporal resolution that can capture seasonal-scale variations (Casado et al., 2023).

Several studies have reported the empirical linear relationship between the isotopic composition of snow and local temperature (Dansgaard, W., 1964; Lorius et al., 1969; Lorius and Merlivat, 1977; Touzeau et al., 2016).

Nevertheless, this relationship is not stable in time and space, as widely documented by direct snow observations and model
results (Goursaud et al., 2018; Jouzel, 1997). These variations are related to changes in evaporative conditions and transport pathways to condensation sites which influence the isotopic equilibrium and kinetic fractionation processes at each step of the distillation trajectories (Charles et al., 1994; Masson-Delmotte et al., 2011; Werner et al., 2011, Casado et al., 2017). In addition, the second order parameter deuterium excess ($d$-$excess$ = $\delta$D – 8* $\delta^{18}$O, Dansgaard, W., 1964) is highly sensitive to kinetic effects occurring both during evaporation at the ocean surface and along distillation pathway during atmospheric
transport. As a result, while over the ocean $d$-$excess$ is very sensitive to near-surface relative humidity and weakly to temperature, it exhibits the opposite behaviour in central East Antarctica, where it may reflect, at least partially, different moisture origins. (Petit et al., 1991; Risi, 2013; Vimeux, 1999; Jouzel, 2003; Masson-Delmotte et al., 2005; Neumann, 2005). However, the reliability of ice core $d$-$excess$ in tracking air mass trajectories is debated due to the local influence of post-depositional processes in the snowpack. These processes can significantly alter the snow isotopic composition during
prolonged exposure at the atmosphere-snow interface and modify the original precipitation's isotope-temperature relationship (Petit, 1982; Stenni et al. 2016, Touzeau et al., 2016, Casado et al., 2018). Key post-depositional mechanisms include sublimation, water vapor exchange, condensation, wind-driven snow redistribution, and vapor diffusion within the snowpack (Casado et al., 2021, 2018; Wahl et al., 2022; Ollivier et al., 2025). Among these, sublimation plays a particularly important role during the warmer months, as it can significantly lower d-$excess$ values (Landais et al, 2017, Casado et al, 2021).
These post-depositional alterations are poorly understood and often overlooked in climatic reconstructions, underscoring the need to better constrain the processes shaping the isotope-temperature relationship in surface snow to improve the reliability of ice core-based temperature estimates (Xiao et al., 2013; Ma et al., 2020).



In this study, we investigate the isotopic composition of surface snow samples collected roughly every 20 km during the East Antarctic International Ice Sheet Traverse (EAIIST - Traversa et al., 2023) in 2019-2020. This scientific traverse extended from the coast in Adélie Land into the high interior of the East Antarctic Plateau. Along this route, we identify a marked transitional zone in moisture origin between the Indian and Pacific Oceans that fell over the past decade. We assess the influence of these distinct sources on the spatial variability of the relationship between $\delta^{18}O$ composition of snow and the 2 m air temperature. In addition, we present the *d-excess* vs $\delta^{18}O$ relationship, showing the different supersaturation pathways observed in cold conditions. To provide a broader context, we extend our analysis to the previous Antarctic snow dataset compiled by Masson-Delmotte et al., (2008). A comparison between observations and simulated water stable isotopes of snow precipitation, through the atmospheric general circulation model LMDZ6iso confirms the model's ability to correctly predict the $\delta^{18}O$-temperature spatial slopes. Subsequently, the model's use in further defining temporal slopes based on the variation of moisture origins - crucial for paleoclimatic reconstructions from ice cores - is investigated as well. Finally, the surface snow samples collected along the plateau during the outward and return ways of the traverse allow assessment of post-depositional alteration over time. By comparing these samples with model outputs representing the isotopic composition of precipitation unaffected by such processes, we can isolate the impact of post-depositional effects on the isotopic composition of the snow during summer months. This quantification is achieved by applying a metamorphism model proposed by Casado et al. 2021.

## 2. Methods

### 2.1. Geographical area EAIIST

The EAIIST traverse took place during the austral summer 2019-2020. The traverse started in coastal Adélie Land, near the high accumulation site of Dumont D'Urville (DDU: 66°39' S, 140°01' E; elevation = 47 m a.s.l.; mean annual temperature of -10.8°C), until the interior plateau site of Megadune (MD: 80°35' S, 121°35' E; elevation = 2973 m a.s.l), and then returned along the same route back to DDU. The traverse route passed by the Dome C site (DC: 75°06' S, 123°23' E; elevation = 3233 m a.s.l.; mean annual temperature of -54.5°C) where the EPICA ice core was drilled and where the Concordia station is currently located. Dome C represents the maximum elevation of the traverse. Although the traverse is topographically divided at Dome C, the transition between the predominance of Indian and Pacific air mass origins occurs farther south (see Results).

### 2.2. Snow Samples and Isotope Measurements

Snow samples were collected approximately every 20 km over 1,600 km route. Furthermore, the traverse included the collection of additional surface snow samples during the return journey MD to Dome C. Two types of surface snow samples were collected: 85 surface samples (top 3 cm of snow) and 52 bulk samples corresponding to snow integrated from a vertically dug 1 m deep snowpit. The samples were collected with 50 mL Corning tubes which were sealed to prevent air exchange and kept frozen until arrival to laboratories in Europe. The samples were distributed between the Laboratoire des Sciences du Climat et de l'Environnement (LSCE) and Ca' Foscari University of Venice (UNIVE) for water isotopic analysis. The samples



were melted only immediately prior to measurement to minimise potential alterations of the water isotopic composition.

Analyses were performed using a Cavity-Ring Down Spectroscopy (CRDS) analyser PICARRO model L2130-i at UNIVE and LSCE. The isotopic composition of snow is expressed in delta-notation (‰) (Craig, 1961) relative to laboratory standards, which were previous calibrated against the international standards V-SMOW (Vienna Standard Mean Ocean Water) and V-SLAP (Vienna Standard Light Antarctic Precipitation). The measurement precision of the Picarro instrument is equal to 0.1 ‰ for $\delta^{18}$O and 1.0 ‰ for $\delta$D.


To estimate the order of magnitude of spatial isotopic variability in surface snow at one location, we report the standard deviation (SD) from previous Antarctic datasets. At Dome C (Casado et al., 2018), the local variability is around 3.4 ‰ for $\delta^{18}$O and 4.1 ‰ for *d-excess* (defined as 2 SD of replicates obtained in an area of 100 x 100 m). During the EAIIST traverse, surface samples were obtained by mixing snow from 10 locations randomly selected over a similar area of 100 x 100 m,

representing an average. Therefore, the impact of spatial variability on the error of the mean is expressed as the standard deviation of the mean (also known as standard error, SE):

1)      $SE = \frac{s}{\sqrt{n}}$

where *s* is the SD calculated from individual samples and *n* is the number of subsamples. Applying this, the expected spatial variability for our composite surface samples is 3.4 ‰ / √(10) ~1.1 ‰ for $\delta^{18}$O and 4.1 ‰ / √(10) ~1.3 ‰ for *d-excess*. Similar

SE are observed in the coastal area of Dumont D'Urville, based on the top 2.5 cm of snowpit samples reported by Landais et al., (2017). Overall, while these values can vary across sites separated by hundreds of kilometres, we set an indicative uncertainty arising from local spatial variability of 1.1 ‰ for $\delta^{18}$O and 1.3 ‰ for *d-excess*, as reference for interpreting our data.

To provide a broader spatial analysis, we compare our data with the Antarctic surface snow database of Masson-Delmotte et al., (2008), which offers a comprehensive overview of isotopic variability across the continent. From the original dataset, which includes different types of snow samples, we selected surface snow, bulk snow, snowpit and firn cores that capture signal ranging from annual to approximately 20 years, based on sample depth and local accumulation rates.

## 2.3. FLEXPART Back-trajectories

The Lagragian particle dispersion model FLEXPART (FLEXible PARTicle) is employed to calculate 10-day back-trajectories of air masses at 12-hour temporal resolution for each sampling site of the traverse located on the Antarctic Plateau. These trajectories are derived for the 500-hPa pressure level, representing mid-tropospheric transport pathways reaching the plateau. With two trajectories calculated per day over the 2009–2019 period, we computed approximately 7,300 trajectories for each site. To allow for a comprehensive multi-annual evaluation and emphasise the air masses leading precipitation events at the

sampling sites, the trajectories were averaged and weighted by ERA5 precipitation rates (see Section 2.4). The classification





of air mass origins into Indian, Pacific and Atlantic sectors is based on their latitudinal and longitudinal distributions, as illustrated in Section 3.2, Fig. 2.

## 2.4. ERA5 Climatic signal

We use ERA5 reanalysis data (provided by the European Centre for Medium-Range Weather Forecasts) to estimate the
precipitation interval and associated temperature conditions related to snow accumulation of surface and bulk snow samples (Fig. 1b). ERA5 provides hourly precipitation and temperature values at 0.25° spatial resolution.

Due to decreasing accumulation rates from the coast (100-300 mm weq yr$^{-1}$) to inland sites (20-50 mm weq yr$^{-1}$), the surface samples near DDU represent a couple of months accumulation compared to up to 6 months of accumulation on the plateau. Bulk samples reflect ~1 year near the coast and up to 15 years on the plateau. This estimate considers only snowfall-driven
accumulation - excluding effects from wind redistribution, erosion, or sublimation—but provides a distinction between seasonal (surface) and multi-seasonal signals (bulk) signals recorded in the samples.

To investigate the $\delta^{18}O$-temperature ($\delta^{18}O$-T) relationship, we compare two temperature metrics. The first is the average 2 m temperature (T) which mainly captures the spatial climatic variability across the continent. The second is the precipitation-weighted temperature ($T_{pw}$) which has a more temporal emphasis reflecting the thermal conditions during snowfall events.
Indeed, $T_{pw}$ is an average temperature where the values are weighted according to the amount of precipitation during each time interval. We calculate T and $T_{pw}$ for each sampling site, based on the accumulation interval corresponding to surface snow samples (see Section 3.1). Furthermore, we calculate the average multi-annual (1980-2020) temperature for both EAIIST and Antarctic snow isotopic database (Masson-Delmotte et al., 2008), enabling a consistent spatial analysis for samples collected during different campaigns.

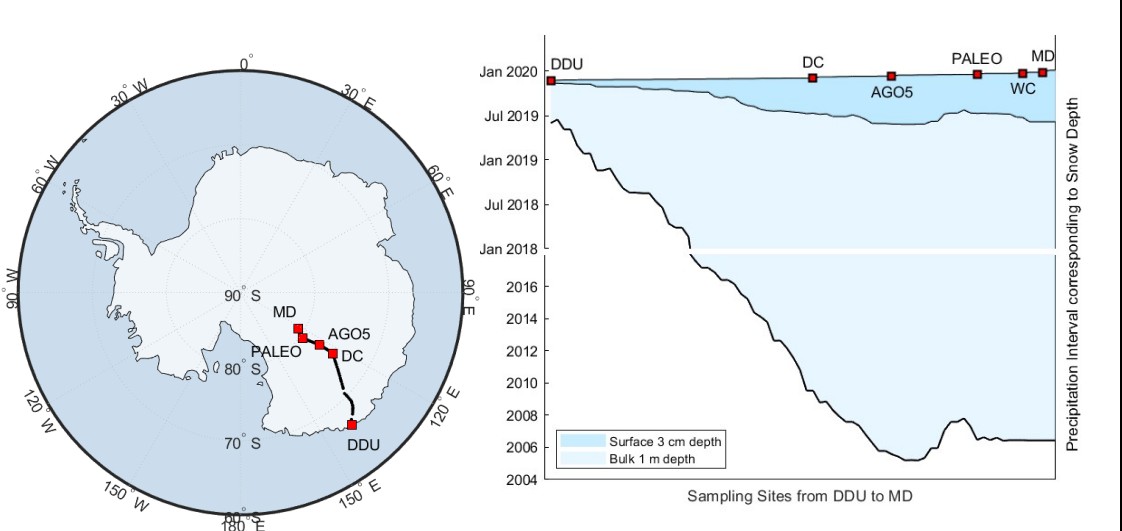




**Figure 1. (left) Map of EAIIST traverse with principal sampling sites. (right) Accumulation period (from ERA5) corresponding to surface and bulk snow samples across the traverse.**

## 2.5. LMDZ6iso Model Data

LMDZ6iso (Risi et al., 2010) is the isotope-enabled version of the atmospheric general circulation model (AGCM) LMDZ6
(Hourdin et al., 2020). We use the LMDZ6iso version 20231022.trunk with the NPv6.3 physical package (Hourdin et al., 2023), which is nearly identical to the IPSL-CM6A atmospheric setup (Boucher et al., 2020), used for phase 6 of the Coupled Model Intercomparison Project CMIP6 (Eyring et al., 2016). We use LMDZ6's standard horizontal Low Resolution (LR) longitude-latitude grid (144×142), which corresponds to a 2.0° resolution in longitude and 1.67 ° in latitude. The vertical grid comprises 79 levels, with the lowest atmospheric level approximately 7 m above ground level (AGL) at Dome C. The
simulation is nudged towards 6-hourly three-dimensional fields of temperature and wind from the ERA5 reanalysis (Hersbach et al., 2020), using a relaxation time scale of 3 hours. Nudging is excluded below the sigma-pressure level equivalent to 850 hPa above sea level, allowing the physics and dynamics of the model to operate freely within the boundary layer. Surface ocean boundary conditions are derived from ERA5 reanalysis monthly mean sea surface temperature and sea-ice concentration fields. The simulation used in this study is described and evaluated over Antarctica in Dutrievoz et al., (2025). Snow samples
are simulated by stacking precipitation events until reaching the target thickness, corresponding to surface and bulk sample depths. The snow layers are precipitation-weighted before averaging, ensuring consistency when compared with the surface and bulk snow data.

## 2.6. Snow metamorphism model

The snow metamorphism model proposed by Casado et al., (2021) describes the relative isotopic variations in snow induced
by different processes. Additionally, the model quantifies the flux of water transferred from the snow to the atmosphere, starting from the isotopic composition of surface snow remaining after sublimation R$_{sub-snow}$, as described the following Eq. 2:

$$2) \qquad R_{sub\ snow}^{i} = \alpha_{eq-sub}^{i} \left[ R_{snow}^{i} \left( \frac{D^i}{D} \right)^{k} (1 - RH) + RH \cdot R_{a}^{i} \right]$$

Where R$_{snow}$ is the initial composition of the snow, D$^i$ and D are the cinetic diffusivity coeffients, $\alpha_{eq-sub}$ is the fractionation coefficient related to sublimation, RH is the relative humidity (Merlivat and Jouzel, 1979) and R$_a$ is the isotopic composition
of atmospheric vapor. The exponent k is the roughness parameter (Craig and Gordon, 1965) , here set to 0.4. In particular, the model predicts that sublimation during the warmest months lead to a *d-excess* / $\delta^{18}$O slope steeper than -2 ‰ / ‰, whereas precipitation input follows a slope of -0.5 ‰ / ‰.

The isotopic difference between R$_{snow}$ and R$_{sub\ snow}$ (expressed as ratio) is used to compute the flux of water transfered from the snow to the atmosphere. This computation takes into account the fractionation coefficient associated to sublimation and
the initial water quantity of the snow sample. This approach will be used in section 4.2, to isolate the impact of sublimation in surface snow, comparing the outbourn and return samples of the traverse. In addition, we simulate the final composition of the





snow, starting from the outboud values and incorporating the snow precipitation imput from LMDZ6iso model and taking into account the contribution of sublimation effects for the number of free-precipitation days.

## 3. Results

### 3.1. Air mass back-trajectories

We present precipitation-weighted air mass back-trajectories over the decadal period 2009-2019 for four representative plateau sites (Fig. 2). At Dome C, 100 % of the back-trajectories originate from the Indian Ocean (red), as expected for this part of the East Antarctic Plateau (Sodemann and Stohl, 2009). At AGO5, approximately 70 % of precipitation events are associated with the Indian sector, with a secondary influence from Pacific Ocean (blue). Southern sites like PALEO and MD, are influenced by the Pacific Ocean, which accounts for over 85 % of precipitation events, with minor contributions from the Indian (5%) and Atlantic (10%) sectors. Based on these observations and the findings of Sodemann and Stohl, (2009) we set the transition between predominately Indian-sourced to predominately Pacific-sourced air masses at approximately 78°S along the EAIIST route (see Appendix A, Fig.A1).

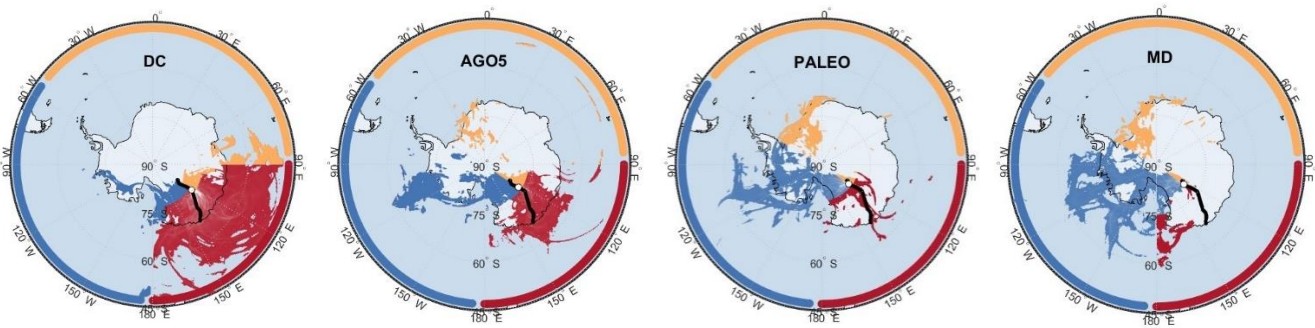

**Figure 2. Precipitation-weighted air mass back-trajectory distributions for the period 2009–2019 at four key sites: Dome C, AGO5, PALEO, and MD. Colors indicate air mass origin, with red, blue and orange representing the Indian, the Pacific and the Atlantic Ocean sector, respectively.**

### 3.2. ERA5 Temperature signal

The $\delta^{18}O$ of surface samples (top 3 cm) collected during the outbound and return ways of the traverse, which represent a time interval of few months, exhibits strong correlation with both T and $T_{pw}$ ($R^2$=0.94). However, the two relationships yield different slopes of 0.9 ‰ °C$^{-1}$ and 0.6 ‰ °C$^{-1}$, respectively (Fig. 3).

The $\delta^{18}O$-T slope is consistent with the value obtained from surface, bulk and shallow snow samples of the Antarctic database (Masson-Delmotte et al., 2008) for Adélie Land based on multi-annual temperature (1980-2020). This similarity validates the use of multi-annual temperature (referred as T in the next Sections) for spatial analysis of snow samples that represent at least





an annual signal. On the other hand, the $\delta^{18}$O-T$_{pw}$ slope is very close to that observed for precipitation at Dome C (0.5 ‰ °C$^{-1}$ - Dreossi et al., (2024)), suggesting that the $\delta^{18}$O-T$_{pw}$ relationship is closer to approximating the temporal relationship between snow isotopes and local weather.

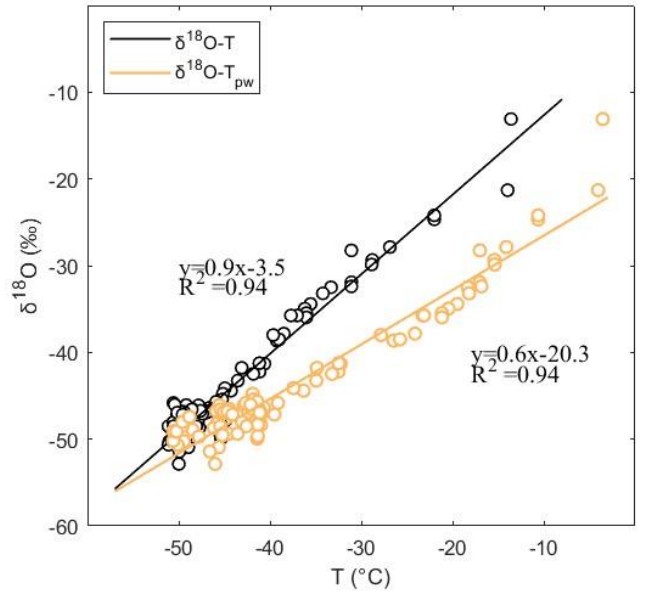


**Figure 3. $\delta^{18}$O-T vs $\delta^{18}$O-T$_{pw}$ relationships evaluated for surface snow samples, calculating mean temperature and precipitation-**
**weighted temperature over time intervals corresponding to 3 cm of snow accumulation.**

### 3.3. Spatial Distribution of Bulk Isotopic Composition

The snow isotopic composition of bulk samples ranges from -20.0 ‰ at the coastal site near DDU to -52.5 ‰ at the most inland Megadune site (a comparison with surface samples is provided in Appendix B, Fig. B1). This range greatly exceeds the uncertainty expected from spatial variability per site (~1.1 ‰, see Methods Section 2.2), and these values fall well within

the range of previous Antarctic surface snow isotopic composition (Fig. 4). Between DDU and Dome C, temperature and $\delta^{18}$O both decline in parallel as the altitude increases. However, south of Dome C toward Megadune, the temperature increases as the altitude decreases, but $\delta^{18}$O slightly decreases. Similarly, the *d-excess* increases linearly from DDU to Dome C but then remains constant further south. Following our back-trajectory air mass results, we divide the EAIIST snow isotopic dataset into two sections split at 78°S.

Section 1 (north of 78°S) exhibits the expected linear relationship between isotopic composition of snow and local climatic and geographic parameters (Tab. 1, Fig. 5). These factors include distance from the nearest coast, latitude, elevation and mean annual 2 m air temperature. In contrast, in Section 2 (south of 78°S), these linear correlations disappear for all the mentioned parameters.



Within Section 2, where no abrupt temperature variations are observed, two anomalous sites are notable: one located
20 km south of PALEO site, with a $\delta^{18}O$ value of -47.9‰, and another at the Windcrust site (WC) with a value of -48.9‰,
roughly 4 ‰ higher than those of the nearby samples. Both sites are also characterised by extremely low *d-excess* values of
7.0‰ and 3.3‰, respectively, compared to those observed on the plateau. Such values likely indicate local sublimation
(Casado et al., 2021; Wahl et al., 2022). Supporting this interpretation, field observations indicate the presence of large hoar
crystals at these sites which experience relatively large strong wind, potentially of katabatic origin (Bintanja et al., 1998).
Although weaker on the plateau than along the continental margins, these winds can persists for extended periods, creating dry
conditions that promote sublimation during summer (Grazioli et al., 2017). In Section 3.6 and 4.2, we further investigate the
impact of sublimation on surface snow based on the comparison between the samples collected during the outward and return
legs of the traverse.

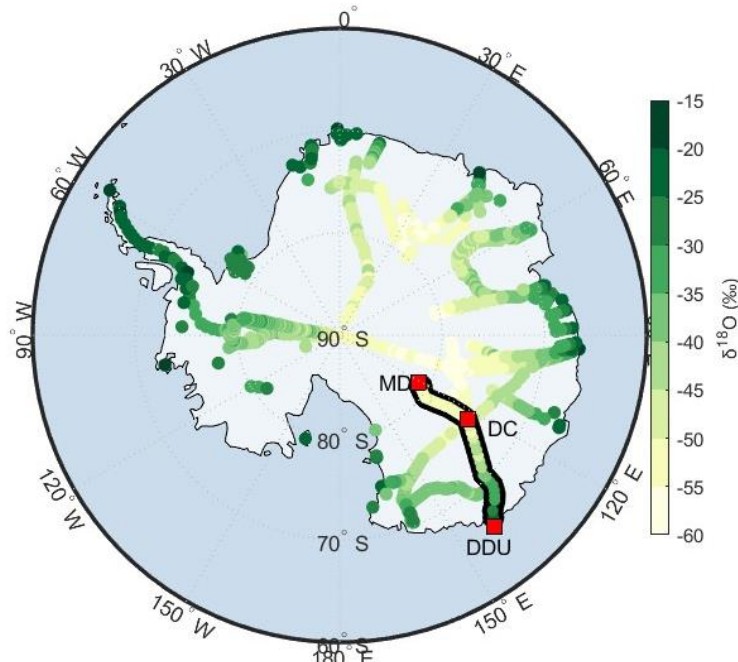

**Figure 4. Map of Antarctica showing the distribution of the $\delta^{18}O$ values of EAIIST bulk samples (marked in black) along with surface, bulk and shallow snow samples from the Antarctic dataset (Masson-Delmotte et al., 2008).**

**Table 1. Slope and correlation coefficients ($R^2$) of linear relationships between isotopic composition and geographical/climatic variables, calculated for the Antarctic dataset and Sections 1 and 2 of the EAIIST traverse. Relationship with $R^2 > 0.5$ are shown in bold.**

| | | Distance from coast | | Sin of latitude | | Elevation | | Temperature | |
|---|---|---|---|---|---|---|---|---|---|
| | | Slope ‰ (100 km)$^{-1}$ | $R^2$ | Slope ‰ (°)$^{-1}$ | $R^2$ | Slope ‰ (100 m)$^{-1}$ | $R^2$ | Slope ‰ °C$^{-1}$ | $R^2$ |
| **Antarctic database** | $\delta^{18}O$ | -2.1 | 0.71 | 0.57 | 0.00 | -1.0 | 0.76 | 0.9 | 0.90 |
| | dxs | 0.9 | 0.61 | -0.83 | 0.01 | (above 2000 m) 0.4 | 0.50 | (below -45°) -0.35 | 0.52 |





| | | | | | | | | |
|---|---|---|---|---|---|---|---|---|
| **EAIIST Section 1** (north 78°S) | $\delta^{18}O$ | **-2.5** | **0.92** | **496.7** | **0.95** | **-1.3** | **0.80** | **0.98** | **0.95** |
| | dxs | **0.8** | **0.75** | **-165.21** | **0.75** | (above 2000 m)**0.9** | **0.57** | (below -45°)**-0.28** | **0.54** |
| **EAIIST Section 2** (south 78°S) | $\delta^{18}O$ | -0.04 | 0.00 | 12.0 | 0.00 | 0.36 | 0.00 | -0.16 | 0.00 |
| | dxs | 0.26 | 0.03 | -74.5 | 0.03 | -0.47 | 0.03 | 0.10 | 0.00 |

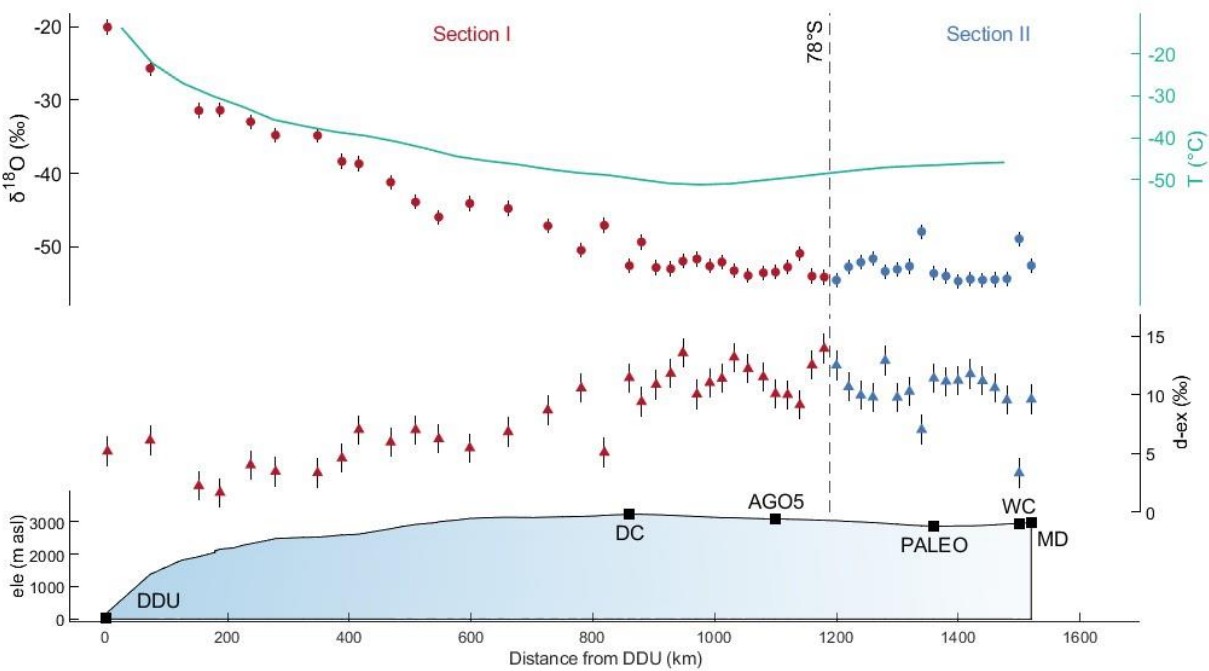

**Figure 5. Spatial distribution of $\delta^{18}O$ and *d-excess* values of bulk samples for Section 1 (red dots – north of 78°S) and Section 2 (blue dots – south of 78°S). The vertical bars represent the uncertainty associated with local variability. Temperature, elevation and distance from DDU are reported.**

### 3.4. Impact of the moisture origins on the isotopic composition

We extend the division between Pacific and Indian sectors to the Antarctic database, based on Sodemann and Stohl, (2009). To do this, we classified the region at west of 60°W and the area near the Ross Sea as Pacific sector, while the sampling sites located north then 80°S and between 60°E and 180°E as Indian sector. For the Antarctic dataset, the linear regressions for the $\delta^{18}O$-T relationship independently calculated for each of the two datasets share a common slope of 0.9 ‰ °C$^{-1}$ but differ in intercept by 7.2 ‰ (Fig. 6 - left). In the *d-excess* vs $\delta^{18}O$ relationship, under cold condition (i.e. for $\delta^{18}O$ values lower than -45 ‰), the two sectors exhibit distinct compositions, with mean slightly higher *d-excess* values (~2 ‰) observed for the Indian samples. Instead, in warmer coastal regions, *d-excess* shows higher variability (Fig. 6 - right).





The comparison with bulk samples of EAIIST reveals that Section 1 - primarily influenced by Indian sector – exhibits isotopic values consistent with those of the corresponding divide of the Antarctic dataset. In contrast, Section 2 shows a pattern more aligned with the behaviour observed for the Pacific sector. These aspects will be discussed further in Section 4.1.

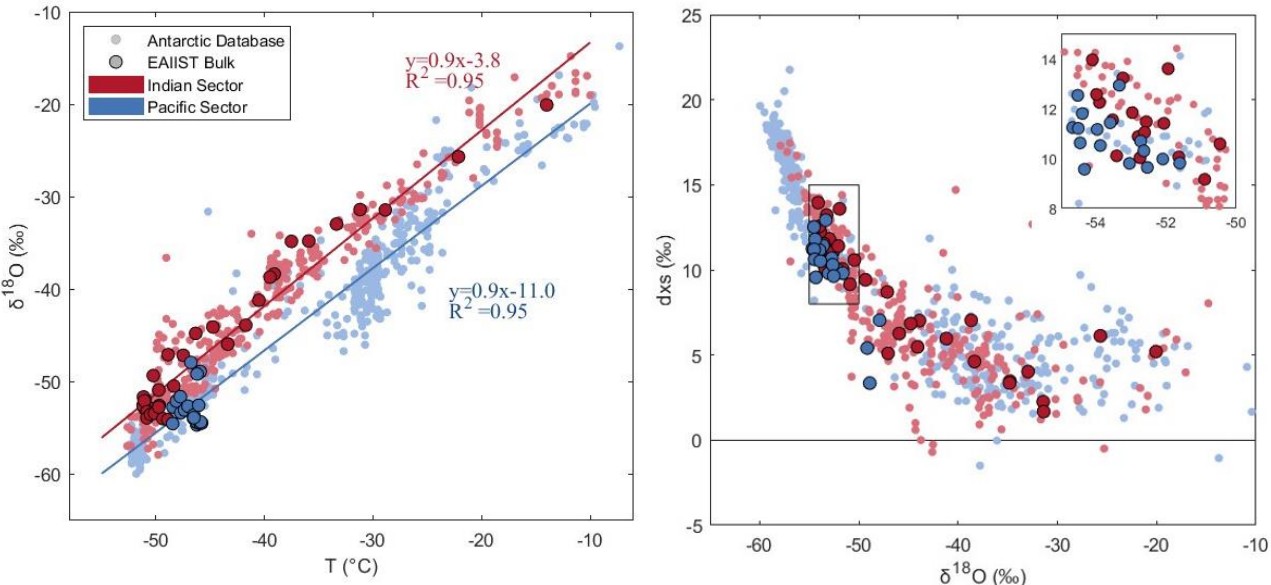


**Figure 6. (left) $\delta^{18}$O-T relationship and (right) *d-excess* vs $\delta^{18}$O for bulk samples of the traverse compared to Antarctic database for Pacific and Indian sectors.**

### 3.5. LMDZ6iso Data Model

Here, we compare modeled $\delta^{18}$O and d-*excess* values for bulk samples simulated by LMDZ6iso with observation (Fig. 7).

For $\delta^{18}$O (Fig. 7 - left), the model successfully predicts the isotopic values, following the different $\delta^{18}$O-T relationship between Indian and Pacific sector as observed in the snow (Appendix C, Fig. C1). For d-excess, the model overestimates the values for both sectors by approximately 8 ‰ (Fig. 7 - right). This offset may be attributed to two main factors. First, *d-excess* in the model is primarily related to the tuning of the supersaturation parameter for isotopes (λ) within the AGCM. The value of λ equalt to 0,004 K$^{-1}$ is selected to achive a optimal compromise to correctly similate both $\delta^{18}$O and *d-excess* in surface snow

(Dutrievoz et al., 2025). Second, post-depositional processes - not accounted by the model – can alter the isotopic composition of snow after deposition. In particular, as mentioned in Section 3.3, sublimation is commonly linked to a reduction in *d-excess* values (Landais et al., 2017; Wahl et al., 2022) that may partially explain the observed difference in the snow. The influence of sublimation, along with the model's performance in reproducing spatial and temporal slopes, will be further discussed in Section 4.






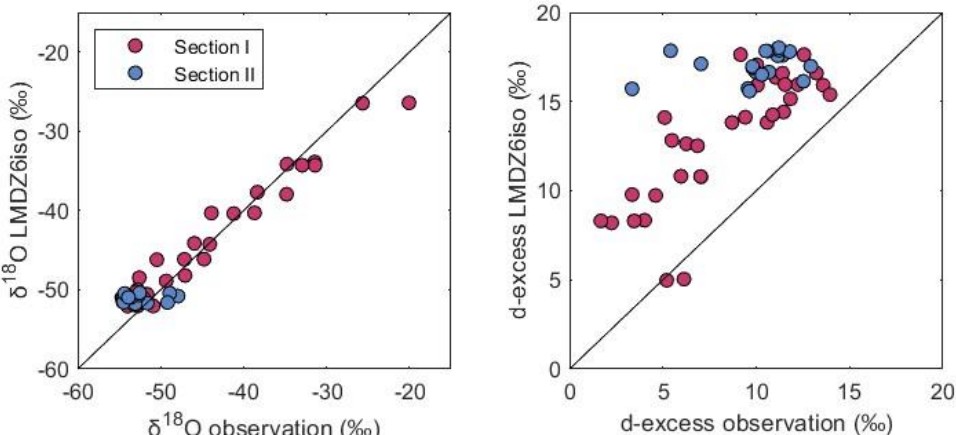

**Figure 7. LMDZ6iso vs bulk observations for (left) $\delta^{18}$O and (right) *d-excess*. The black line is for linear regression.**

### 3.6. Temporal Variability of Surface Isotopic Composition

To assess the impact of post-deposition effects on the isotopic compoition of the snow, we compare surface samples (representing a seasonal signal) collected during the outward and return ways of the EAIIST traverse across the plateau region (sampled between 10 and 50 days apart). The return isotopic composition is indeed on average slightly higher for $\delta^{18}$O, by

approximately 3 ‰ (Fig. 8a), and with a *d-excess* significantly lower, on average 5 to 10‰. Limited snowfall for most of the sites – especially over the plateau – suggests that post-deposition processes could be responsible for the difference observed. Indeed, these conditions allow for prolonged interaction at the snow-atmosphere interface, enhancing the post-depositinal effect that alter the isotopic composition of surface snow.

We evaluate the contribution of precipitation to the change of surface snow between the outward and return journey using the LMDZ6iso simulation outputs. The contribution of precipitation would lead to an increase of up to 2 ‰ for $\delta^{18}$O and a decrease of about 1 ‰ for *d-excess* at sites north of 78°S. South of 78°S, the model suggests no change in surface snow isotopic composition due to precipitation. Using ERA5, we investigate the precipitation patterns between the outward and return samplings. In Section 1, we identify 7-20 precipitation-free days and a total accumulation of 5-8 mm of fresh snow

(corresponding to ~25 % of the surface sample). In contrast, Section 2 experienced 10-20 dry days with no significant precipitation inputs (Fig. 8c).

The post-deposition metamorphism leading to isotopic changes in the snow at the surface-atmosphere interface will be discussed in Section 4.2.



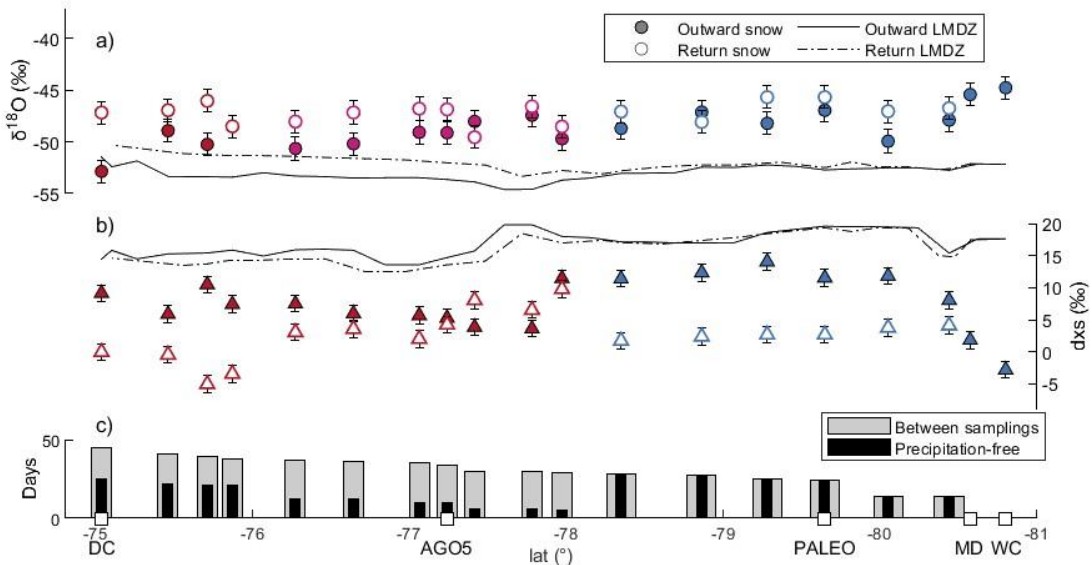

**Figure 8. a) $\delta^{18}$O and b) d-*excess* composition of outward and return surface snow samples (dots) in comparison to the LMDZ6iso model outputs (lines). Vertical bars indicate spatial variability. c) The days between the two journeys (grey) and the precipitation-free days prioir the return sampling (black).**

**4. Discussion**

**4.1. Air Mass Origins and Isotopic Paleothermometer**

Previous studies have investigated past changes in atmospheric circulation under different climatic conditions and hypothesised their influence on the $\delta^{18}$O-T relationship (Reijmer et al., 2002; Schlosser et al., 2004). These studies consistently highlight that variations in moisture source regions, transport pathways, and sea-ice conditions can significantly influence the $\delta^{18}$O–

temperature relationship in Antarctic precipitation. This underscores that stable isotope signals are not solely controlled by local temperature. The EAIIST traverse - crossing a transitional region between Indian and Pacific sector - provide valuable and direct observations of the impact that distinct air mass origins have on the snow isotopic composition in present day.

Our spatial analysis revealed two distinct $\delta^{18}$O-T relationships for the two sectors investigated (Fig.6 - left). Both relationships are characterised by a slope of 0.9 ‰ °C$^{-1}$, reflecting comparable thermal history of air mass from the coasts to

the high-elevation regions in the interior of the continent (Helsen et al., 2007). Yet, the differing y-intercept values suggest





contrasting initial isotopic composition of water vapor, likely reflecting both differences in the source ocean regions and in the specific trajectories followed by air masses before reaching the continent.

As shown in Figure 7a, the isotopic variation for the two sectors is well predicted by the model. These results highlight the model potential not only to reproduce spatial slopes but also to predict temporal slopes (see Section 3.2). To further assess this, we compute monthly precipitation from LMDZ6iso at Dome C, AGO5, PALEO and MD sites – accounting for the moisture origin variations (Fig. 9). A temporal $\delta^{18}$O-T slopes between 0.4 and 0.5 ‰ °C$^{-1}$ (R$^2$= 0.8) are retrieved, consistent for the whole spectrum of trajectories. This slope is similar to the one derived from monthly snow precipitation samples at Concordia Station, Dome C, equal to 0.5 ‰ °C$^{-1}$ (Dreossi et al., 2024).

In the *d-excess* vs $\delta^{18}$O comparison (Fig. 6b), the higher *d-excess* variability in warmer coastal areas (< 2000 m a.s.l.) is linked to its preservation of information about changes in close moisture source. These include temperature and relative humidity in evaporative region, as well as fluctuations in sea ice extent. Conversely, in inland regions (> 2000 m a.s.l.), located at the end of the distillation pathway, the *d-excess* in precipitation becomes more sensitive to condensation temperature, since it is regulated by independent supersaturation functions (Uemura et al., 2012; Stenni et al., 2016; Touzeau et al., 2016; Landais et al., 2017). This behaviour is clearly visible from the Antarctic database and is also slightly evident in the EAIIST data for very cold sites (i.e., $\delta^{18}$O < -50 ‰),  with higher *d-excess* values observed for the Indian sector compared to the Pacific sector for the same $\delta^{18}$O composition.



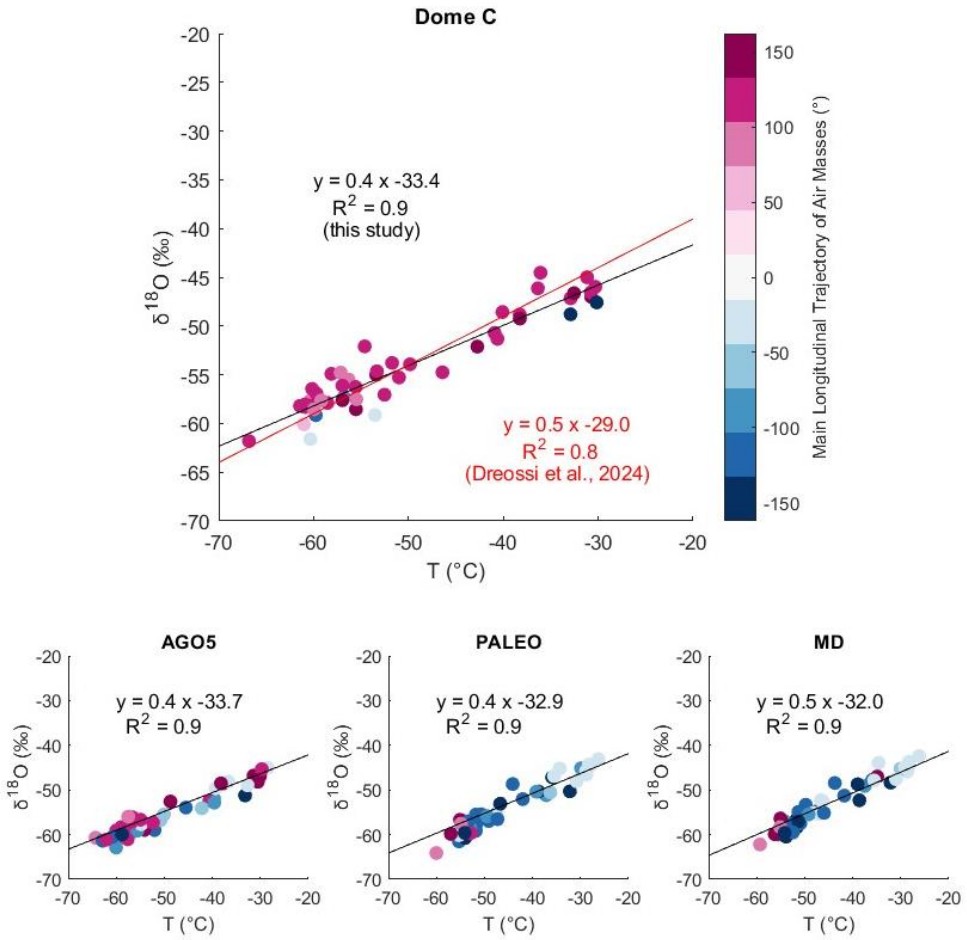

**Figure 9. $\delta^{18}$O-T relationship for monthly snow samples simulated by LMDZ6iso model at the sites Dome C, AGO5, PALEO and MD. The color scale indicates the dominant longitudinal origin of air masses associated with each monthly precipitation event.**

### 4.2. Impact of Sublimation on Surface Snow

In Section 3.6, we showed that precipitation alone could not explain the change of isotopic composition of the surface samples between the outbound and return journeys. Here, we assess whether sublimation - commonly associated with an enrichment in $\delta^{18}$O along with a decrease of *d-excess* (Dietrich et al., 2023; Hughes et al., 2021) - can explain, at least in part, the variations observed between the outbound and return surface snow samples. To this end, we compare the relative variations in *d-excess* and $\delta^{18}$O against the snow metamorphism model proposed by Casado et al., (2021) (see Section 2.6). Overall, our observations support this framework: isotopic changes along Section 2 – during which no precipitation occur according to ERA 5 – closely follow the modelled metamorphic modification of isotopic composition characterised by a slope of *dxs / $\delta^{18}$O* of -2 ‰ / ‰ (blue dots in Fig. 10) (Casado et al., 2021). For Section 1, as shown in Section 3.6, precipitation accounts for a 25% of the resurn sample weight, as a result, we expect a mix between the contibution of precipitation, characterised by a slope *dxs / $\delta^{18}$O*





of -0.5 ‰ / ‰ (Casado et al., 2021). We observe that the surface snow in Section 1 evolved following a slope of -1.1 ‰ / ‰ (red dots in Fig. 10), which match the hypothesis of an input dominated by sublimation, with a contribution from the precipitation input.

We use Section 2 data, driven solely by sublimation, to estimate the flux of water transfered from the snow to the atmosphere (eq. Section 2.6). The resulting sublimation fluxes range from 0.04 to 0.09 mm weq day$^{-1}$ (Fig. 10), consistent with values reported by Ollivier et. al., (2025), who observed sublimation-driven vapor fluxes ranging from 0.05 to 0.35 mm weq day$^{-1}$ during summer at Dome C. We use this isolated sublimation impact for modeling the final composition of the snow along the whole traverse, using outward snow samples as the initial condition. This calculation accounts for the number of precipitation-
free days between samplings (from ERA5), and incorporates precipitation input from the LMDZ6iso model. The final $\delta^{18}$O and *d-excess* modeled composition is estimated as:

$$\text{Final Snow}_{model} = \text{Initial Snow}_{obs.} + \text{Precipitation Input}_{LMDZ6iso} + (\text{Sublimation} \times N°_{dry\ days})$$

When accounting for sublimation effects, the predicted isotopic values showed better agreement with the observed variability in the return snow samples (Fig. 11) compared to the simulation from the LMDZ6iso model alone (Fig. 8). These findings
highlight the signficant impact of sublimation – even if limited – on the isotopic composition of surface snow in this region.

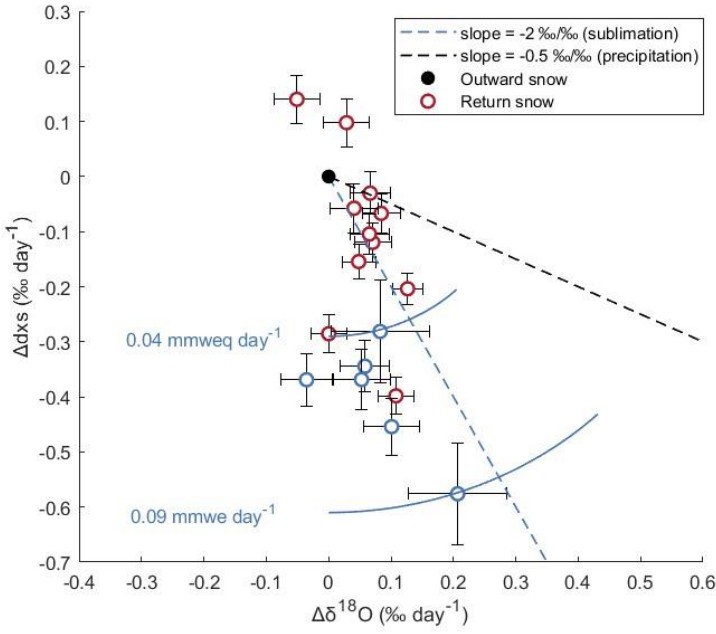

**Figure 10.** *d-excess* **vs** $\delta^{18}$**O evolution of snow per day for Sector I (empty red dots) and II (empty blue dots) centered on outward**
**composition (black dot), in comparison with modeled dxs/$\delta^{18}$O sublimation slope (blue dashed line) and precipitation slope (black**





dashed line). The fluxes relative to sublimation are expressed in mm weq day⁻¹. The errorbars represent the uncertainty from spatial variability.

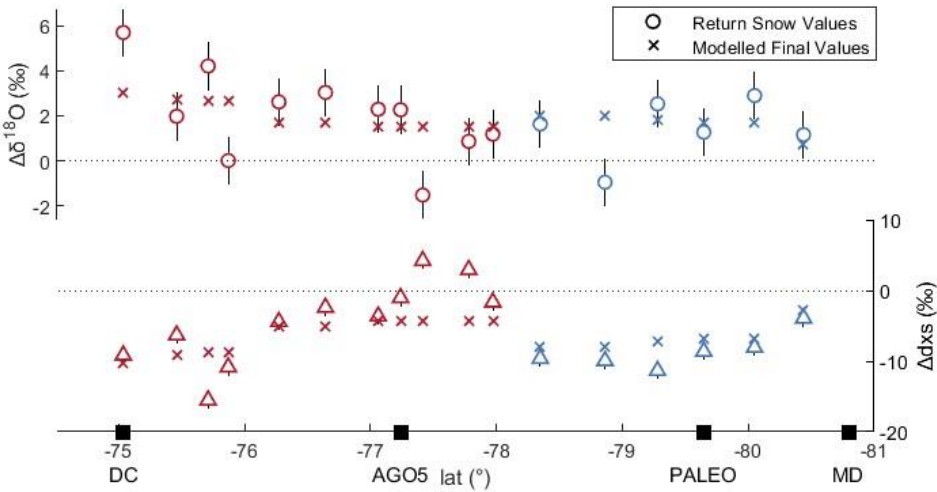


**Figure 11. Differences in $\delta^{18}O$ (top) and *d-excess* (bottom) between outward snow samples and (i) return snow values, and (ii) modelled final snow calculated based on both the precipitations input (LMDZ6iso) and the sublimation input. Vertical bars indicate the spatial variability of snow samples.**

## 5. Conclusions

In this study, we presented new isotopic data from surface and bulk snow samples collected along the EAIIST traverse, which covers a remote and previously unexplored region of the East Antarctic Plateau. The traverse crosses a transitional zone between two dominant air mass origins: the Pacific and Indian Oceans. By investigating the $\delta^{18}O$-T and d-excess vs $\delta^{18}O$ relationships, we identified distinct isotopic signature associated with each sector, highlighting the influence of moisture sources on the snow isotopic composition. These findings were further supported by a spatial comparison with the broader

Antarctic snow dataset of Masson-Delmotte et al., (2008).

The EAIIST dataset also confirmed the ability of the LMDZ6iso model to adequately reproduce the spatial distribution of $\delta^{18}O$ in surface snow, accounting for the different moisture origins. Yet, the model showed reduced capability in simulating the *d-excess* values. This limitation can be attributed primarily to model parametrization and secondarily to post-depositional effects, such as sublimation, which significantly affect the *d-excess* signal. The impact of sublimation was assessed by comparing

surface snow samples collected during the outbound and return ways of the traverse across the plateau – where low accumulation rate and strong snow drift lead to prolonged summer atmosphere-snow exposure, enhancing this effect. Assuming 20 precipitation-free days per summer season, the associated post-depositional variation could induce an enrichment



of +2 ‰ yr$^{-1}$ in $\delta^{18}$O and a decrease of up to -8 % yr$^{-1}$ in *d-excess* (based on an avearge methamorisphism of +0.1 ‰ day$^{-1}$ and -0.4 ‰ day$^{-1}$ during summer, see Section 4.6).


In conclusions, these results provided key insights for the future interpretation of ice cores collected along the EAIIST traverse. (i) Surface and bulk snow samples defined the range of isotopic variability linked to distinct moisture sources, which must be considered in regions where multiple air mass trajectories coexist and vary over time. (ii) The comparison between observations and LMDZ6iso outputs confirmed the model's ability to capture both spatial and temporal $\delta^{18}$O-T slopes. In

particular, the temporal slope of 0.5 ‰ °C$^{-1}$ – previously established from precipitations at Dome C (Dreossi et al., 2024) and commonly used to calibrate the isotopic records from this site – is validated for all the sites in the EAIIST transition region. This supports its use for calibrating the $\delta^{18}$O-T paleothermometer also in areas lacking direct observations in time. (iii) While the observed annual scale variation attributed to sublimation is smaller than the inter-sector $\delta^{18}$O differences (Pacific vs Indian origins), it represents a non-negligible factor for accurate reconstruction of past temperature. (iv) Lastly, the finding that

relatively close drilling sites (e.g. Dome C and PALEO, separated by only few hundred kilometres) can experience different precipitation pathways, could be used as a strategy to enhance the temperature signal retrieved, combining ice cores that archive the same climatic signal via distinct precipitation events.





**Appendix**

**Appendix A: Londitudinal distribution of air mass orising on the East Antarctic Plateau sampling sites**

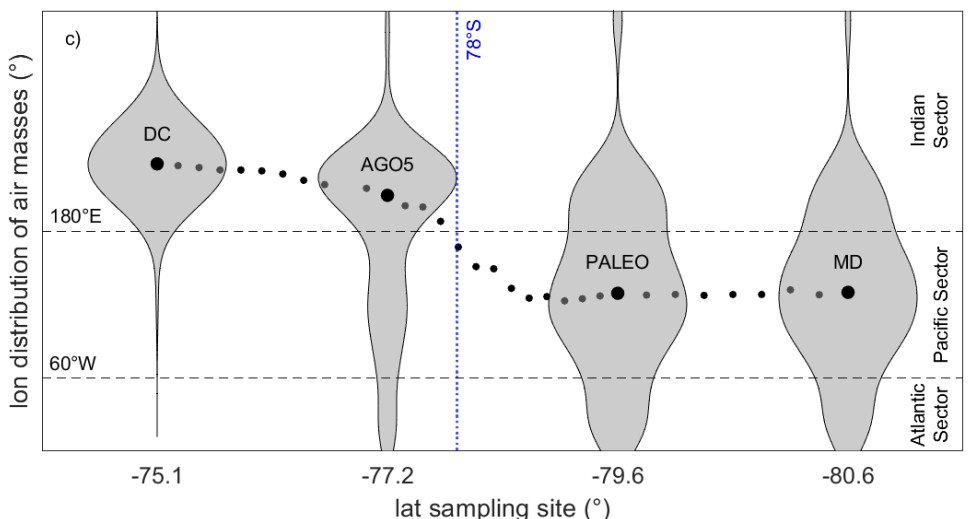

**Figure A1. Longitudinal distribution back-trajectories over the 2009–2019 period for four representative sites: Dome C, AGO5,**
**PALEO, and MD. The median longitudinal origin of air masses for the 33 sampling sites across the Antarctic Plateau is indicated**
**by black dots.**



**Appendix B: Spatial distribution of surface and bulk samples**

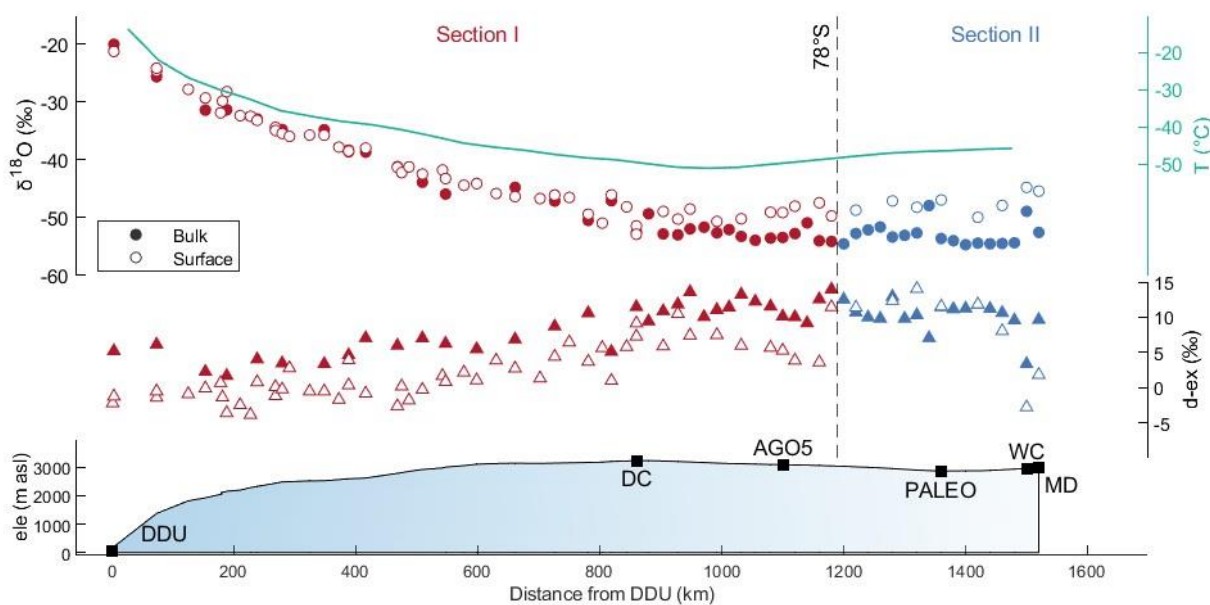


**Figure B1. Spatial distribution of *δ¹⁸O* and *d-excess* values of surface (empty markers) and bulk (filled markers) samples for Section 1 (red– north of 78°S) and Section 2 (blue– south of 78°S). Temperature, elevation and distance from DDU are reported.**

**Appendix C: Comparison between LMDZ6iso bulk predictions and Antartic dataset**





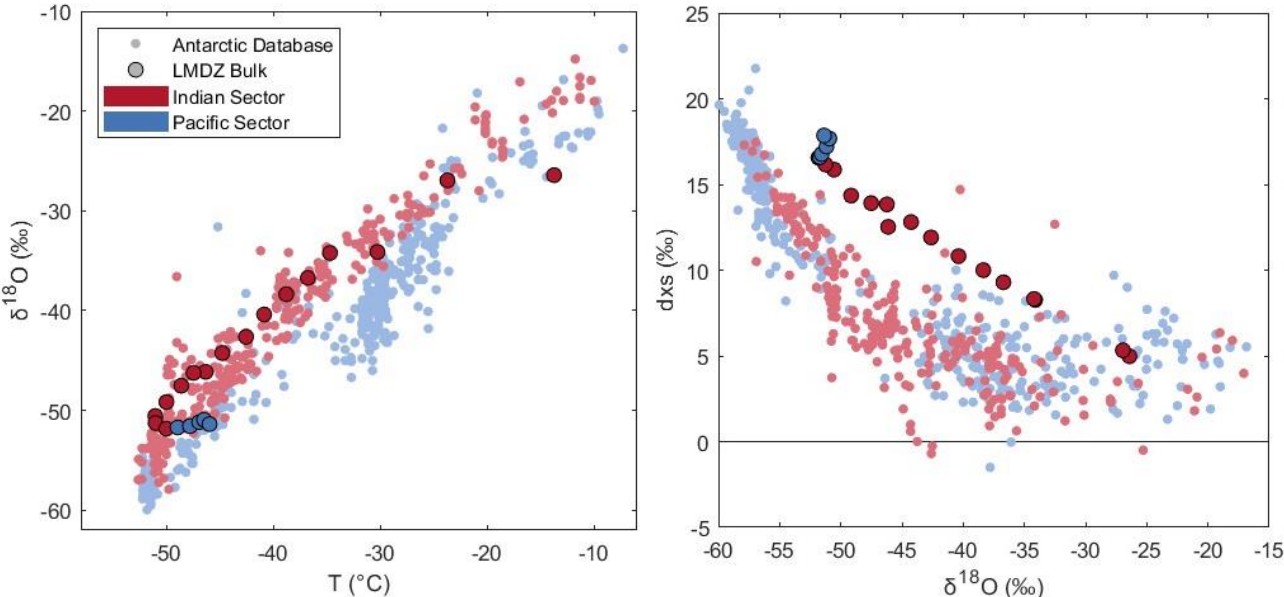


**Figure C1. (left) *δ¹⁸O-T* relationship and (right) *d-excess* vs *δ¹⁸O* for LMDZ6iso bulk samples of the traverse compared to Antarctic database for Pacific and Indian sectors.**



**Author contributions**

AP and MC performed the measurement in Venice and Paris; AP, CLDS conducted the back-trajectory analysis using the
FLEXPART model; AP, ND and CA computed the LMDZ6iso simulations; MC, PDA, JS, AS and MF collected the snow
samples during the EAIIST traverse; AP and MC wrote the manuscript draft; AL, ND, PDA, JS, AS, MF and BS reviewed the
manuscript.

**Competing interests**

The authors declare that they have no conflict of interest.

**Acknowledgements**

The authors would like to thank all the staff of the East International Ice sheet traverse that made possible the collection of the
samples used in this manuscript. We also acknowledge all the logistical support received from the PNRA and IPEV for the
safe samples handling, storage and transportation. This project received funding from the "Programma Nazionale per la Ricerca
in Antartide" ("EAIIST" PNRA16_00049-B and "EAIIST-phase2" PNRA19_00093 projects).

**Financial support**

This work was supported by the Polar Science PhD scholarship from Ca' Foscari University of Venice and the Erasmus+
program. This research has been supported by the European Research Council, Project SAMIR (HORIZON: European
Research Council, grant no. 101116660).




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
