# Peer review of "Air Mass Origin Effects on Antarctic Snow Isotopic Composition: An Observation and Modelling Study"

_EGUsphere, 2025_

## Referee Comment (RC1)

**Review of "Air Mass Origin Effects on Antarctic Snow Isotopic Composition: An Observation and Modelling Study" by Petteni et al.**

The authors present a study analyzing the stable water isotopic composition of snow samples collected in East Antarctica. These measurements are complemented with back-trajectory calculations, ERA5 reanalysis data, and output from two models (LMDZ6iso and a snow metamorphism model). Together, these datasets are used to address the key question of what climatic information is preserved in the stable water isotopic composition of firn and ice cores, particularly in low-accumulation regions such as the East Antarctic Plateau. This question has been addressed by a number of studies in the previous years highlighting the relevance of the topic presented in this manuscript. The manuscript fits into the aims and objectives of the EGUsphere, but I suggest revisions before publication.

Overall Comments

The study addresses two major aspects: i) the analysis of different air mass origins, and ii) the influence of post-depositional effects, particularly sublimation, on the isotopic signal of surface snow. However, only the first aspect is reflected in the current title. I recommend revising the title so that both themes are represented equally.

Overall, the manuscript is generally well written but would benefit from thorough proofreading before final submission. Figure labelling is inconsistent: in several cases, panels are labelled "left" and "right" in the captions but referred to as "a" and "b" in the text. Please adopt a consistent style throughout the manuscript (e.g., "a, b, c..."). In addition, several figure captions lack sufficient detail. For example, Figure 1 does not explain the abbreviations for the locations shown on the left panel, and the right panel is missing proper axis labels (is the x-axis in kilometres?). Please also cite any mapping software used (e.g., Quantarctica).

In the Results section, some passages read more like discussion, while in the Discussion section new results and figures are introduced. I recommend ensuring that results and discussion are clearly separated.

At multiple points, the manuscript states that "significant" differences or impacts are observed, but no explanation is given as to how significance was determined. Please clarify the methods used to assess significance and discuss the findings in relation to uncertainties. Similarly, model uncertainty is not considered when comparing observations with model output (e.g., Fig. 11).

Figure 11 underpins many of the conclusions, but several issues remain:
- Model uncertainty is not addressed.
- ERA5 precipitation uncertainty, which is well known, is not discussed.
- The "better agreement" between the return-sample variability and the LMDZ6iso output is claimed but not quantified; moreover, the modelled $\delta^{18}O$ values show no variability.

- o Wind-driven redistribution, which is a key process on the East Antarctic Plateau, is not mentioned. This process could substantially influence accumulation patterns and isotopic signatures and should be discussed alongside sublimation.

I agree with the authors that sublimation may have a strong effect on the isotopic composition of snow. However, other depositional and post-depositional processes, especially wind redistribution, should not be neglected. I recommend expanding the discussion to include these processes and their potential impact on the results.

**Specific comments (minor)**

I only have two minor comments to the abstract.
- In l.17, it is written that depositional and post-depositional effects lead to large unceratinties in the use of stable water isotopes as proxy in Antarctica. The unceratinties of d18O as a proxy is strongly depending on the specific location in Antarctica with East Antarctica or low accumualtion regions having a much larger uncertainty than high accumulation areas in West Antarctica for instance. Please be more specific here.
- Secondly, when mentioning that LMDZ6iso captures the spatial variations accuratley, it would be nice to see a number showing how well the model captures the variability. It will strengthen the statement.

- l. 42f.: One sentence is not a full paragraph. Please incorpoarte this sentence into the following paragraph.
- L. 49: pleae add an s → distilation pathways
- L. 56f.: What is the difference between sublimation, water vapor and vapor diffusion?
- L. 66: I don't understand "that fell over the past decade". What do you mean here?
- L. 70: are you using all data from Masson-Delmotte (2008) or only a subset (e.g. the East Antarctic Plateau)?
- L. 77: please add brakets → Casado et al. (2021)
- Chapter 2.2:
  - o Please carefully check the language in this chapter.
  - o Please provide more details on the snow samples.Did you take several samples at a location or only one each time? Did you always take a surface and a bulk sample at the same location? Why do you have 85 surface samples but only 52 bulk samples?
  - o We know that different labs show discrepancies when measuring the same samples. Have you performed an independent quality controll or something similar between both labs?
  - o What is your uncertainty for d-excess values? It would be interesting to have a number for d-excess as well, not only $\delta^{18}O$ and $\delta D$, to compare this to d-excess variations.
- Chapter 2.4:

- - l. 130: I don't agree to call the output of ERA5 *snow accumulation*. Considering the mentioned depositional and post-depositional modifications, I would refer to *snowfall* or *snow precipitation* provided by ERA5.
  - L. 133: can you be more specific what you mean with *couple of months*?
- L. 161f.: Are you considering densification for the bulk samples? This might be relevant for the bulk samples on the plateau, considering that they contain up to 15 years of snowfall.
- L. 167: cinetic → kinetic
- L. 176f.: outburn/outboud → outbound; free-precipitation → precipitation-free
- L. 182: Looking at Fig. 2 the DC plot, I also see lines that are colored in blue and orange. Would that imply that not all 100% are originating from the Indian Ocean?
- L. 195: few → please be more specific if possible
- Table 1: can you mark Section 1 and 2 in the map of Fig.4 ? Are relationships with $R^2 > 0.5$ tested for significance?
- L. 254: Did you test for significant differences?
- L. 267ff.: For me, this reads already like discussion. You can consider to move this part to the discussion in Section 4.
- L. 285: how did you test for significance?
- L. 326 and Fig. 9: All plots with the new data show an $R^2$ of 0.9 but 0.8 is mentioned in the text. Please correct this.

---

## Author Comment (AC1)

Dear editor,

We would like to thank the two reviewers for their comments. We have been working toward a new version of the manuscript taking their respective comments into account. We include and number the reviewers' comments in black. The referees' comments have been addressed individually, as requested by the journal.
Our responses are in blue, and the modifications to the manuscript in red in this response file.

On behalf of all the co-authors,
Agnese Petteni

**Reviewer #1:**

Review of "Air Mass Origin Effects on Antarctic Snow Isotopic Composition: An Observation and Modelling Study" by Petteni et al.

The authors present a study analyzing the stable water isotopic composition of snow samples collected in East Antarctica. These measurements are complemented with back-trajectory calculations, ERA5 reanalysis data, and output from two models (LMDZ6iso and a snow metamorphism model). Together, these datasets are used to address the key question of what climatic information is preserved in the stable water isotopic composition of firn and ice cores, particularly in low-accumulation regions such as the East Antarctic Plateau. This question has been addressed by a number of studies in the previous years highlighting the relevance of the topic presented in this manuscript. The manuscript fits into the aims and objectives of the EGUsphere, but I suggest revisions before publication.

Overall Comments

1) The study addresses two major aspects: i) the analysis of different air mass origins, and ii) the influence of post-depositional effects, particularly sublimation, on the isotopic signal of surface snow. However, only the first aspect is reflected in the current title. I recommend revising the title so that both themes are represented equally.

We agree with the referee, and we adapt the title as follow:
"Air mass origin and local impacts on Antarctic snow isotopic composition: an observation and modelling study"

2) Overall, the manuscript is generally well written but would benefit from thorough proofreading before final submission. Figure labelling is inconsistent: in several cases, panels are labelled "left" and "right" in the captions but referred to as "a" and "b" in the text. Please adopt a consistent style throughout the manuscript (e.g., "a, b, c…").

We have corrected the text and accordingly adjusted the figures, as suggested by the referee.

3) In addition, several figure captions lack sufficient detail. For example, Figure 1 does not explain the abbreviations for the locations shown on the left panel, and the right panel is missing proper axis labels (is the x-axis in kilometres?). Please also cite any mapping software used (e.g., Quantarctica).

We have corrected the figure as suggested by the referee. We included a statement at the end of the manuscript in the acknowledgement to state that the maps were plotted with Matlab using the mapping toolbox.

4) In the Results section, some passages read more like discussion, while in the Discussion section new results and figures are introduced. I recommend ensuring that results and discussion are clearly separated.

We agree with the referee's comment. We have moved sections that were previously in the Discussion into the Results, and transferred all paragraphs related to the discussion of the results into the two Discussion subsections concerning air mass origins and sublimation. This has improved the clarity of the two parts.

5) At multiple points, the manuscript states that "significant" differences or impacts are observed, but no explanation is given as to how significance was determined. Please clarify the methods used to assess significance and discuss the findings in relation to uncertainties. Similarly, model uncertainty is not considered when comparing observations with model output (e.g., Fig. 11).

The referee is correct. We have revised the sentences where we previously mentioned a "significant" difference or impact, indicating the p values < 0.05 (Pearson correlation).

Regarding model uncertainty, we have addressed this point by adding a new Fig. 10 in the revised manuscript, which provides an improved comparison between observations and model output. In this figure, we explicitly include the LMDZ6iso model uncertainty in predicted values, as reported by Dutrievoz et al. (2025).

When sublimation is included, the mean absolute difference between modelled and observed values decreases for both $\delta^{18}O$ and d-excess, demonstrating an improved agreement with observations when accounting for this process.

We have implemented the results with Fig. 10 and presented it in the Results section:

Line 350: "The difference between observed versus modelled final values are shown in Figure 10. Red symbols represent Section I sampling sites, characterized by high precipitation (> 1 mm w.e.) between outbound and return samplings, for which freshly precipitation represents ~25% of the sampled snow. Blue symbols represent Section II sites with low precipitation (< 1 mm w.e.), characterised by negligible precipitation. The modelled values are presented considering either only the precipitation input or both precipitation and sublimation effects. Including sublimation in the computation reduces the discrepancy for all Section II sites and for the majority of Section I sites. The mean absolute difference decreases from 1.9 to 1.3 ‰ for δ18O , and from 6.6 to 2.9 ‰ for d-excess."

[Figure]

**Figure 10. Difference between return-snow observations and modelled $\delta^{18}O$ and d-excess values, for Section I (red) and Section II (blue). Modelled values are calculated considering either precipitation only (crosses) or both precipitation and sublimation effects (circles). Error bars represent the uncertainty of the LMDZ6iso model in simulating the isotopic precipitation at Concordia (Dutrievoz et al., 2025). The x-axis indicates the cumulative precipitation between the outward and return samplings.**

6) Figure 11 underpins many of the conclusions, but several issues remain:
   o Model uncertainty is not addressed.

This aspect has been addressed in point 5).

   o ERA5 precipitation uncertainty, which is well known, is not discussed.

We agree with the referee on the importance of discussing the uncertainty in ERA5, and we have added this point in Section 4.3.

Line 439: "Including this sublimation effect in the modelled isotopic predictions substantially improved the agreement with observations, reducing the discrepancy for d-excess compared to simulations considering precipitation input alone (Fig. 10).

This improvement is particularly evident at sites where cumulative precipitation between samplings is negligible (< 1 mm w.e.). The remaining differences and variability between observed and modelled values can be partialy attributed to uncertainties in ERA5 precipitation and LMDZ6iso model. Previous studies have shown ERA5 overestimates precipitation over the East Antarctic Plateau, with biases reaching up to 50% relative to satellite-based measurements (Roussel et al., 2020). As a result, our modelling likely represents the maximum contribution of precipitation, implying that the metamorphism would be even greater if actual precipitation were lower. We emphasize that the aim of this study is not to quantify the ability of the combined ERA5-LMDZ6iso in reproduce the absolute isotopic values, but rather to evaluate whether accountig sublimation improved the qualitative representation of surface snow isotopic composition compared to precipitation-only scenarios. These results further reinforced the key role of post-depositional processes in shaping the isotopic composition of surface snow."

  o The "better agreement" between the return-sample variability and the LMDZ6iso output is claimed but not quantified; moreover, the modelled $\delta^{18}O$ values show no variability.

This aspect has been addressed in point 5).

  o Wind-driven redistribution, which is a key process on the East Antarctic Plateau, is not mentioned. This process could substantially influence accumulation patterns and isotopic signatures and should be discussed alongside sublimation.

This aspect has been addressed in point 7).

7) I agree with the authors that sublimation may have a strong effect on the isotopic composition of snow. However, other depositional and post-depositional processes, especially wind redistribution, should not be neglected. I recommend expanding the discussion to include these processes and their potential impact on the results.

We agree with the referee, we included in the revised version of the manuscript:

Line 426: "On the Antarctic plateau, post-depositional effects mainly include wind-driven snow redistribution, and sublimation.

Snow transport by wind is a relatively local process, typically mixing snow from the surrounding areas and one of the main contributions to stratigraphic noise (Hirsch et al., 2023). Studies suggest that the snow shuffled by wind redistribution can reach distances up to ~100 km (Scarchilli et al., 2010; Frezzotti et al., 2007). Such mixing generates stratigraphic effect between precipitation events, leading to $\delta^{18}O$ variability of up to 4.4 ‰ within the uppermost 6 cm at Kohnen Station in the plateau interior (Münch et al., 2016). In our dataset, this effect is reduced because we mixed surface snow collected over an extended area at each sampling site and by integrating 1 m of snow depth for bulk samples.

To quantify sublimation, mainly impacting the snow metamorphism in summer, …"

Specific comments (minor)
8)   on the specific location in Antarctica with East Antarctica or low accumulation regions having a much larger uncertainty than high accumulation areas in West Antarctica for instance.

Please be more specific here.

We agree with the referee's comment, and we have implemented the manuscript as follow:
Line 17: "The magnitude of these uncertainties strongly depends on site location, with larger impacts in low-accumulation regions of East Antarctic Plateau."

9) • Secondly, when mentioning that LMDZ6iso captures the spatial variations accurately, it would be nice to see a number showing how well the model captures the variability. It will strengthen the statement.

We agree with the reviewer on the importance of clarifying how the model predicts isotopic composition. In the spatial analysis, we now explicitly state that the LMDZ6iso model distinguishes two different δ18O–temperature relationships for the Indian and Pacific sectors. Then, for the modelled values that also include sublimation impact, we provide the differences between observations and modelled values.

Line 26: "Comparison with LMDZ6iso simulations indicates that the model successfully captures the spatial variability of $\delta^{18}$O-temperature relationship between different basins, with statistically significant correlations ($p < 0.05$) when the analysis is extended to the Antarctic dataset. This agreement further suggests the model's ability to predict the temporal slope required for calibrating isotopic ice-core records used for temperature reconstructions, even in regions influenced by multiple moisture sources. Temporal slopes based on monthly precipitation values range from 0.4 to 0.5 ‰ °C⁻¹ for the EAIIST drilling sites. Finally, we quantify the impact of sublimation on isotopic composition of surface snow. Including sublimation in the modelling of surface snow reduces the discrepancy between observed and modelled values, compared to simulations accounting precipitation, from 1.9 to 1.3 ‰ for $\delta^{18}$O and from 6.6 to 2.9 ‰ for d-excess. These results highlight the key role of this post-depositional process on the Antarctic Plateau."

10) • l. 42f.: One sentence is not a full paragraph. Please incorporate this sentence into the following paragraph.

Taken into account

11) • L. 49: please add an s → distillation pathways

Taken into account

12) • L. 56f.: What is the difference between sublimation, water vapor and vapor diffusion?

The referee pointed out that the sentence was ambiguous. We have revised it to improve clarity:
Line 60: "Key post-depositional mechanisms include wind-driven snow redistribution and vapor exchange with the atmosphere, such as sublimation-condensation processes, and vapor diffusion within the snowpack driven by forced ventilation (Steen-Larsen et al., 2014; Casado et al., 2021, 2018; Wahl et al., 2022; Ollivier, 2025)."

13) • L. 66: I don't understand "that fell over the past decade". What do you mean here?

We have adjusted the text following this comment. In the revised version, this sentence has been removed from the introduction, and the corresponding results are now presented directly in the Results section.

14) • L. 70: are you using all data from Masson-Delmotte (2008) or only a subset (e.g. the East Antarctic Plateau)?

Taken into account and implemented:
Line 128: "To provide a broader spatial analysis, we compare our data with the Antarctic surface snow database of Masson-Delmotte et al., (2008), which offers a comprehensive overview of isotopic variability across the continent. From the original dataset, which includes different types of snow samples, we selected surface snow, bulk snow, snowpit and firn cores that capture signal ranging from annual to approximately 20 years, based on sample depth and local precipitation rates. For the comparison with our snow samples, we divide the dataset in Pacific and Indian sectors, based on Sodemann and Stohl, (2009). To do this, we classified the region at west of 60°W and the area near the Ross Sea as Pacific sector, while the sampling sites located north then 80°S and between 60°E and 180°E as Indian sector."

15) • L. 77: please add brackets → Casado et al. (2021)
Taken into account

• Chapter 2.2:
  16) o Please carefully check the language in this chapter.
Taken into account
  o Please provide more details on the snow samples. Did you take several samples at a location or only one each time? Did you always take a surface and a bulk sample at the same location? Why do you have 85 surface samples but only 52 bulk samples?
Taken into account
Line 94: "Two types of surface snow samples were collected: 85 surface samples, representing the upper 3 cm of snow, and 52 bulk samples, consisting of snow integrated over a vertically dug 1 m-deep snowpit. Surface samples were taken at each stop during daytime approximately every 20 km along the 1,600 km route from DDU to MD, and onward to DC. Bulk sampling required longer processing time and was therefore carried out only during lunch and evening stops"

  17) o We know that different labs show discrepancies when measuring the same samples. Have you performed an independent quality control or something similar between both labs?
Taken into account
Line 110:" Previous inter-calibration experiments revealed mean discrepancy between UNIVE and LSCE measurements of the same samples equal to 0.14 ‰ and 0.80 ‰, for $\delta^{18}O$ and d-excess respectively (Petteni et al., 2025)."

  18) o What is your uncertainty for d-excess values? It would be interesting to have a number for d-excess as well, not only $\delta^{18}O$ and $\delta D$, to compare this to d-excess variations.
Line 106: "The accuracy of PICARRO measurements was determined as the mean difference between measured and true values of laboratory standards, with uncertainty represented by their standard deviation. This yielding an accuracy of -0.01 ‰ for $\delta^{18}O$, -0.07 ‰ for $\delta D$, and -0.02 ‰ for d-excess, with corresponding uncertainties of ±0.07 ‰, ±0.4 ‰, and ±0.4 ‰."

• Chapter 2.4:
  19) o l. 130: I don't agree to call the output of ERA5 *snow accumulation*.
    Considering the mentioned depositional and post-depositional modifications, I would refer to *snowfall* or *snow precipitation* provided by ERA5.
This has been taken into account and modified accordingly throughout all sections.

  20) o L. 133: can you be more specific what you mean with *couple of months*?
Taken into account
Line 155:"represent from 1 to 3 months of snowfalls"

  21) • L. 161f.: Are you considering densification for the bulk samples? This might be relevant for the bulk samples on the plateau, considering that they contain up to 15 years of snowfall.
We used density values reported by Ooms et al. (2025), which show limited densification (Fig. A2). The mean densities derived from trench measurements at Dome C are 290 kg m⁻³ for the upper 3 cm of snow and 320 kg m⁻³ for the upper 1 m.
The density values were not reported in the original manuscript and is now included as follows:

Line 146: "Prior to the comparison with ERA5, all snow samples were converted to water equivalent using density values of trench measurements at Dome C (Ooms et al. 2025). The densities are equal to 290 kg m⁻³ for the upper 3 cm of snow and 320 kg m⁻³ for the upper 1 m. For each sample, precipitation events in ERA5 were sequentially accumulated until the target water-equivalent thickness of the sample was reached. Due to the strong gradient in precipitation rates from the coast

(100–300 mm w.e. yr$^{-1}$) to the plateau (20–50 mm w.e. yr$^{-1}$), surface samples near DDU represent from 1 to 3 months of snowfalls, whereas those collected on the plateau correspond to up to ~6 months. Similarly, bulk samples represent approximately 1 year of precipitation in coastal areas and up to 15 years at the highest-elevation sites."

22) • L. 167: cinetic → kinetic
Taken into account

23) • L. 176f.: outburn/outbound → outbound; free-precipitation → precipitation-free
Taken into account

24) • L. 182: Looking at Fig. 2 the DC plot, I also see lines that are colored in blue and orange. Would that imply that not all 100% are originating from the Indian Ocean?
The referee is right; we have corrected the text accordingly.
Line 211: "At Dome C, 90 % of the back-trajectories originate from the Indian Ocean (red), as expected for this part of the East Antarctic Plateau (Sodemann and Stohl, 2009)."

25) • L. 195: few → please be more specific if possible
Taken into account

26) • Table 1: can you mark Section 1 and 2 in the map of Fig.4 ? Are relationships with R2 > 0.5 tested for significance?
Section 1 and Section 2 are now indicated in the figure caption, as suggested. We did not mark the sections directly on the figure, as it is already quite crowded.
The referee is correct that we did not explain how we define statistical significance. R² values greater than 0.5 are shown in bold when p < 0.05 (Pearson correlation).
We have corrected the Table 1 caption as follows:
Line 270: "Table 1. Slope and correlation coefficients (R2) of linear relationships between isotopic composition and geographical/climatic variables, calculated for the Antarctic dataset and Sections 1 and 2 of the EAIIST traverse. Relationships are shown in bold when statistically significant (defined by p < 0.05)."

27) • L. 254: Did you test for significant differences?
No, the difference between the two regression lines was not tested for statistical significance; we only highlighted here the difference between their intercepts.

28) • L. 267R.: For me, this reads already like discussion. You can consider to move this part to the discussion in Section 4.
We agree with the referee's comment. We have reorganized the Results and Discussion sections accordingly.

29) • L. 285: how did you test for significance?
The referee is right. We didn't test the difference statistically, we rephrase the sentence as follow to be consistent with the results showed:
Line 289: "The return isotopic composition is, on average, slightly higher for δ$^{18}$O by approximately 3 ‰ (Fig. 8a), while d-excess is lower, by 5-10 ‰ (Fig. 8b)."

30) • L. 326 and Fig. 9: All plots with the new data show an R2 of 0.9 but 0.8 is mentioned in the text. Please correct this.
Taken into account

**References:**

Frezzotti, M., Urbini, S., Proposito, M., Scarchilli, C., Gandolfi, S., 2007. Spatial and temporal variability of surface mass balance near Talos Dome, East Antarctica. J. Geophys. Res. Earth Surf. 112, F02032. https://doi.org/10.1029/2006JF000638

Hirsch, N., Zuhr, A., Münch, T., Hörhold, M., Freitag, J., Dallmayr, R., Laepple, T., 2023. Stratigraphic noise and its potential drivers across the plateau of Dronning Maud Land, East Antarctica. The Cryosphere 17, 4207–4221. https://doi.org/10.5194/tc-17-4207-2023

Ooms, A., Casado, M., Picard, G., Arnaud, L., Hörhold, M., Spolaor, A., Traversi, R., Savarino, J., Ginot, P., Akers, P., Twarloh, B., Masson-Delmotte, V., 2025. Inter-annual snow accumulation and meter-scale variability from trench measurements at Dome C, Antarctica. EGUsphere 2025, 1–39. https://doi.org/10.5194/egusphere-2025-3259

Petteni, A., Fourré, E., Gautier, E., Spagnesi, A., Jacob, R., Akers, P.D., Zannoni, D., Gabrieli, J., Jossoud, O., Prié, F., Landais, A., Tcheng, T., Stenni, B., Savarino, J., Ginot, P., Casado, M., 2025. Interlaboratory comparison of continuous flow analysis (CFA) systems for high-resolution water isotope measurements in ice cores. Atmospheric Measurement Techniques 18, 5435–5455. https://doi.org/10.5194/amt-18-5435-2025

Scarchilli, C., Frezzotti, M., Grigioni, P., De Silvestri, L., Agnoletto, L., Dolci, S., 2010. Extraordinary blowing snow transport events in East Antarctica. Clim. Dyn. 34, 1195–1206. https://doi.org/10.1007/s00382-009-0601-4

Roussel, M.-L., Lemonnier, F., Genthon, C., and Krinner, G., 2020. Brief communication: Evaluating Antarctic precipitation in ERA5 and CMIP6 against CloudSat observations. *The Cryosphere* 14, 2715–2727. https://doi.org/10.5194/tc-14-2715-2020